# Propagating the Rosary in the Early Qing—A Case Study of del Rosario's Comprehensive Manuscript

## Hongfan Yang

College of Foreign Languages, Fujian Normal University, Fuzhou 350007, China; therese.yanghf@hotmail.com

**Abstract:** Studies on the Rosary in the late Ming and early Qing usually focus on works written by Jesuits and mostly stem from an artistic aspect. This article, however, shifts the focus to *The True Peace of Humankind*, a manuscript written by the Dominican missionary Arcadio del Rosario in the seventeenth century, the first comprehensive book on the Rosary in Chinese. It first summarizes the early-stage propagation of the Rosary in China by the Jesuits and then examines the structure and content of *The True Peace of Humankind*. It is noteworthy that the manuscript repeatedly uses an analogy with flowers to highlight Mary's intercession. Dating back to the Catholic tradition in Europe, the propagation of the Rosary through analogy with flowers resembles the propagation of reciting Buddhist prayers in Chinese society. This article applies contextual studies to explore two main questions: how is del Rosario's manuscript different from the previous texts on the Rosary written by the Jesuits? What is the significance of the manuscript in the context of the Chinese Rites Controversy?

**Keywords:** the Rosary; *The True Peace of Humankind*; Arcadio del Rosario; Yan Mo

## 1. Introduction

For hundreds of years, reciting the Rosary has been a most popular practice among Chinese Catholics. Ever since the late Ming, missionaries had begun to propagate the Rosary in Chinese Christian texts. Yet, studies on the Rosary in Chinese usually focus on the works written by Jesuits in the seventeenth century. Moreover, the examination is mainly from an artistic aspect, with an emphasis on the illustrations in the *Regulation for Reciting the Beads* (*Song nianzhu guicheng* 誦念珠規程) (Qu 2012; Ji 2016; Lopes 2018; Wang 2021). In light of the existent studies, this article shifts the focus to a manuscript on the Rosary written by the Dominican missionary Arcadio del Rosario (Ou Jialüe 歐加略 O.P., 1641–1686).

From their entry into mainland China in the 1630s to the early eighteenth century, the mendicants wrote fewer Chinese works than the Jesuits. Moreover, most of their works were about catechism, prayers, confraternity rules, and hagiography, in contrast with a greater variety of subjects in the Jesuit works (Hsia 2011, pp. 221, 225). At the same time, the Chinese Rites Controversy, which originated from the adaptations to the Chinese tradition of Matteo Ricci (Li Madou 利瑪竇 S.J., 1552–1610), became even more heated and culminated in the condemnation of traditional Chinese rites by papal decree in 1704 (Standaert 2001, pp. 680–81). Problems discussed during the Controversy can be categorized into three areas: questions of how to translate important terms, such as the name of God; questions over whether Christians should participate in sacrifices to Confucius; and questions about Christians' participation in community festivals in honor of non-Christian divinities, whether mass could be said to non-Christian ancestors, etc. (Standaert 2001, pp. 680, 683). In light of this categorization, how to propagate the Rosary may be considered the third area of questions.

This article applies contextual studies to explore two main questions: how is del Rosario's manuscript different from the previous texts on the Rosary written by the Jesuits? What is the significance of the manuscript in the context of the Chinese Rites Controversy? The first section summarizes the early-stage propagation of the Rosary in China

by the Jesuits; the second section examines the structure and content of *The True Peace of Humankind* (*Renlei zhen'an* 人類真安, ca. 1680) from a textual aspect; and the third section explores how Mary's intercession is highlighted through an analogy with flowers in miracle stories recorded in the manuscript, in comparison with a story from Chinese Buddhist texts, as well as similar stories in Chinese Christian texts.

## 2. Early-Stage Propagation of the Rosary in China by the Jesuits

The Rosary is the most popular form of Marian prayer developed in medieval Europe (Fulton 2004, p. 1). In the early China mission, the Jesuit missionaries spread the practice of reciting the Rosary even before the publication of its text. Soon the newly converted Chinese in various regions began to recite this classic Catholic prayer. In the 1580s, the father of Wang Pan 王泮 recited the Rosary every day, kneeling before a picture given by Michele Ruggieri (Luo Mingjian 羅明堅 S.J., 1543–1607) and Ricci (Bernard 1937, part 1, p. 112). In 1606, when João Soerio (Su Ruwang 蘇如望 S.J., 1566–1607) was lying in bed because of illness, more than forty local Chinese Catholics in Nanchang 南昌 assembled to recite the Rosary for their beloved missionary (Bernard 1937, part 3, p. 267).

As the Christian communities developed in the early seventeenth century, the Jesuits began to publish manuals of the Rosary in Chinese. Firstly, this publication belonged to the Jesuit worldwide propagation to make the recitation of the Rosary more common among the faithful (Viller et al. 1987, p. 961). Secondly, it belonged to the Jesuit strategy to evangelize through books in China. The books circulated to places where the missionaries could not go or rarely visited, assisting local Christian communities to grow. Moreover, because of the severe gender segregation in Chinese society, books were a convenient means for the missionaries to preach and teach upper-class women (Amsler 2018, p. 53). As in Europe, early editions of the Rosary in Chinese were manuals to guide Catholics in reciting prayer, generally called "Regulation for Reciting the Beads" (*nianzhu guicheng* 念珠規程), with an emphasis on meditation.

The *Regulation for Reciting the Beads* (*Song nianzhu guicheng* 誦念珠規程, Jap.Sin. I, 43.b, ARSI) is the first manual of the Rosary in Chinese. Recent studies have confirmed that two early editions of Regulation for Reciting the Beads correspond to the same text written by Gaspar Ferreira (Fei Qigui 費奇規 S.J., 1571–1649) (Ji 2016, pp. 38–42). Fu Ji suggests that the first edition was published in Hangzhou 杭州 in 1628 (Ji 2016, pp. 43–46), while Wang Xiliang presumes that it was published during Ferreira's stay in Jiangxi 江西 around 1630 (Wang 2021, p. 96). Paul Brunner, however, holds that the Jesuits would not have waited until 1628 to compose such an essential text (Brunner 1964, p. 39).

The seventeenth century witnessed further propagation of the Rosary in Christian Chinese texts. *The Regulation for Reciting the Beads* was included in the *Daily Services of the Holy Teaching* (*Shengjiao rike* 聖教日課, 1665, BnF Chinois 7353–7354), retitled as the *Fifteen Points in the Rose Prayer to the Holy Mother* (*Shengmu meigui jing shiwu duan* 聖母玫瑰經十五端) (Ferreira 1665, f. 31). In the table of contents, the title is slightly different, written as the *Prayer of Fifteen Points of the Rose to the Holy Mother* (*Shengmu meigui shiwu duan jing* 聖母玫瑰十五端經). In the *Evidence of the Holy Teaching* (*Shengjiao xinzheng* 聖教信證, ca. 1680), the title is shortened to *Fifteen Points in the Rose Prayer* (*Meigui jing shiwu duan* 玫瑰經十五端) and attributed to Ferreira (Han and Zhang 1984, f. 6, p. 305). In late Ming and early Qing, the Rosary was usually called the Prayer of the Fifteen Points (*shiwu duan jing* 十五端經), while the Rose Prayer (*meigui jing* 玫瑰經), the term that is the most frequently used today, only appeared occasionally.

In contemporary Europe, various manuals of the Rosary have been published in different languages. In Medieval literature, the Latin word "rosarium" (Rosary) meant a collection or chain of texts in various disciplines, such as philosophy, law, and medicine. In the fifteenth century, the Carthusians referred to "rosarium" as the collections of the meditative recitation of Hail Mary (Viller et al. 1987, p. 943). The Rosary became a term for one type of Catholic meditative recitation, using the beads for counting. Specifically, it refers to the prayer of the Fifteen Mysteries. The Jesuit missionaries translated the Rosary

as "reciting beads" (*nianzhu* 念珠). When referring to a certain type of Rosary, the title of the specific prayer shall add "for our Lord" or "for the Holy Mother"; when referring to the prayer of the Fifteen Mysteries, it can be simply called the prayer of reciting the beads. The historical Chinese terminology of the Rosary resembles the English terminology, with "prayer" referring to each "bead", and "to say a bead" is to offer each prayer (Fulton 2004, p. 5).

The title of the *Regulation for the Beads* reveals how Catholic missionaries borrowed Buddhist terminology to propagate Catholic prayers. It is the Buddhist monks who introduced the practice of reciting prayer (*nianjing* 念經) and the beads for recitation (*nianzhu* 念珠) into Chinese society (Institut Ricci 2001, IV, p. 651). The common practice of chanting the sutra in Chinese society probably prepared Chinese Catholics for their preference for chanting prayers over murmuring (Amsler 2018, p. 201, note 43).

Ferreira translated the Portuguese word "mystery" (*mysterio*) as "point" (*duan* 端). Among the multiple meanings of "duan 端" as a noun, one is "Affair; specifically the affairs, the points of doctrine, the formulas of prayer"; because of its common usage in Christian texts since late Ming, this meaning has evolved into a secondary explanation, "(Catholicism) Decade (of the chaplet); mystery (of the Rosary)" (Institut Ricci 2001, VI, p. 289).[1] It is noteworthy that Ferreira did not use the word "*aoji* 奧跡", the translation for mystery in Chinese Christian texts (cf. Institut Ricci 2001, I, p. 66). It is interesting that the absence of the word "*aoji* 奧跡" in the early editions of the Rosary in Chinese resembles the early Rosary texts in German, where the German word "Geheimnis" and the Latin word "Mysterium" do not appear. In 1480, the statutes of the confraternity in Venice began using "mysteries" (*misteri*) in the Rosary, and its usage became common in Europe (Viller et al. 1987, p. 956). As for the Rosary in Chinese, the character "*duan* 端" has become the fixed translation for the fifteen decades.

Both before and after Ferreira's publication of the *Regulation for Reciting the Beads*, the Jesuits propagated other rosaries as well. In the early seventeenth century, Niccolò Longobardo (Long Huamin 龍華民 S.J., 1565–1654) translated a prayer collection called the *Daily Services of the Holy Teaching* (*Shengjiao rike* 聖教日課, ca. 1602–1619).[2] Two prayers in this book resemble the *Regulation for Reciting the Beads*, i.e., the *Meditation Rules of Reciting Beads for Our Lord* (*Song wuzhu nianzhu moxiang guitiao* 誦吾主念珠默想規條) and the *Meditation Rules of Reciting Beads for the Holy Mother* (*Song shengmu nianzhu moxiang guitiao* 誦聖母念珠默想規條). These two meditation rules of reciting beads were included in the *Prayers of Sufferings* (*Tongku jingji* 痛苦經蹟), a prayer collection published in 1638 (Brunner 1964, pp. 89–90; Longobardo 1984a, pp. 1123–46; 1984b, pp. 1111–22).

According to the Chinese Christian Texts database, Giulio Aleni (Ai Rulüe 艾儒略 S.J., 1582–1649) published the *Images of Fifteen Points in the Holy Mother's Rose Prayer* (*Shengmu meigui jing shiwu duan tuxiang* 聖母玫瑰經十五端圖像, 1637) in Quanzhou 泉州.[3] In the *Evidence of the Holy Teaching*, the title is shortened to the *Images of Fifteen Points* (*Shiwu duan tuxiang* 十五端圖像) in the list of Aleni's work (Han and Zhang 1984, f. 9, p. 312; cf. Pfister 1934, p. 135). In the first half of the seventeenth century, the Jesuit missionaries published and circulated several manuals of the Rosary among Chinese Catholics. Based on their works, it was a Dominican missionary who wrote the first comprehensive book on the Rosary in Chinese.

## 3. The Manuscript on the Rosary by del Rosario

In the late seventeenth century, the Spanish Dominican missionary Arcadio del Rosario wrote a manuscript called *The True Peace of Humankind*. In *The Sino-European Printing Works in China*, this work is briefly recorded without the author's name as "No. 380, 人類真安 *Jen lei tchen ngan.*—*The True Peace of Humankind*, by a Dominican", followed by a note in smaller size, indicating that this is a manuscript preserved at Xujia hui 徐家匯 (*Ms. à Zi Ka-wei*) (Cordier 1901, p. 62, no. 380). There is also a copy housed in the Bodleian Library, Oxford (MS.Chin.d.43).[4] The current article uses a facsimile of the manuscript preserved

at Xujia hui (Shanghai-ZKW, 95525B) and converts all variant Chinese characters to their commonly used form.

Carrying "rosary" (*rosario*) in his name, del Rosario wrote a five-volume manuscript on the Rosary with abundant materials in 175 pages. Although the manuscript does not contain a table of contents, its structure is clear and even—each of the five volumes contains five chapters. The first volume is a summary of the virtue of praying (*zonglun qide gongxing* 總論祈德公性); the second volume is an introduction to the Prayer of the Fifteen Points (*shiwu duan jing shili* 十五端經事理); the third is an explanation of rules to recite the Prayer of the Fifteen Points (*shiwu duan jing guize* 十五端經規則); the fourth records miracle stories about saving the soul by reciting the Prayer of the Fifteen Points (*shiwu duan jing xianji linghun* 十五端經顯蹟靈魂); and the fifth is miracle stories about saving the body by reciting the Prayer of the Fifteen Points (*shiwu duan jing xianji roushen* 十五端經顯蹟肉身).

As the first comprehensive book on the Rosary in Chinese, *The True Peace of Humankind* complements previous manuals by the Jesuits and expounds Mary's intercession more profoundly. Del Rosario explained the three elements (*liao* 料) of the Rosary, i.e., the Sign of the Cross (*Shenghao jing* 聖號經, the Holy Sign's Prayer), Our Father (*Tianzhu jing* 天主經, the Lord of Heaven's Prayer), and Hail Mary (*Shengmu jing* 聖母經, the Holy Mother's Prayer). Just as Ferreira did, del Rosario presented only parts of the prayer instead of the whole content. The rule for each point provides the description of the point, the meditation, and the procedure, "Recite one Our Father, ten Holy Mother's Prayer, and one Glory Be" (念一遍天主經，十遍聖母經，一遍聖三經) (del Rosario 2013, vol. 3, f. 35, p. 153). Although not included as the element of the Rosary, the doxology, i.e., Glory Be (*Shengsan jing* 聖三經, the Holy Trinity's Prayer), is recited at the end of each point. Both the Sign of the Cross and Glory Be had not been mentioned in the previous manuals of the Rosary by the Jesuits.

Similar to the *Regulation for Reciting the Beads*, the manuscript takes the three-step formula of recitation, offering, and petition. In previous manuals, the Jesuits had already urged Chinese Catholics to meditate while reciting rosaries. Del Rosario, furthermore, categorized meditation into two types.

> When people recite the fifteen points, they must meditate on the reason (*shili* 事理) of the fifteen points. There are two types of meditation: either recite while meditating the reason for that point or first meditate and recite after the meditation.
>
> 人要念十五端，當默想十五端事理。默想有兩樣，或隨念，隨想那一端事理；或先默想，想完乃念。 (del Rosario 2013, vol. 3, f. 33, p. 149)

As the manuals of various rosaries exemplify, it is crucial to meditate during the recitation. The Fifteen Mysteries are presented in three strings (*chuan* 串): firstly, the five joyful points (*huanxi wuduan* 歡喜五端); secondly, the five sorrowful points (*tongku wuduan* 痛苦五端); thirdly, the five glorious points (*rongfu wuduan* 榮福五端) (del Rosario 2013, vol. 3, f. 33, p. 149).[5] The division of the three strings derives from Italian Rosary texts. In the 1480s, either the confraternity in Venice or the one in Florence began to divide the fifteen Mysteries into joyful (*gaudioso*), sorrowful (*doloroso*), and glorious (*glorioso*) (Viller et al. 1987, p. 956).[6] The Chinese term "*chuan* 串" indicates both "a string of beads" and "a string of flowers", echoing two meanings of the Latin word *rosarium* (Fulton 2004, pp. 1–2).

The fifteen points of the Rosary are in fact a brief narrative of essential Catholic teachings, from the dogma of the Incarnation, Passion, and Resurrection of Christ to the doctrine of the Assumption of Mary.[7] Because of Chinese grammar, the content in Chinese can be read in the present tense rather than in the past tense, thus showing a more vivid narration.[8] Reciting the Rosary was one way for Chinese Catholics to learn about and review the essential teachings, especially for female Catholics who were rather isolated from social life.[9] It is customary to recite the five joyful points on Monday and Thursday; the five sorrowful points on Tuesday and Friday; and the five glorious points on Wednesday, Saturday, and Sunday (del Rosario 2013, vol. 3, f. 33, p. 149).[10] Through the propagation of the Rosary, Chinese Catholics embraced the custom of reciting the Rosary in the rhythm of the Catholic liturgical calendar.[11]

The second volume of *The True Peace of Humankind* confirmed Mary's intercession by elaborating on how superior (*gao* 高) the Rosary is in five chapters. The first chapter acknowledged two founders of the Rosary, with the Holy Mother Mary being the first and the saintly Father Dominic (Duo-ming-wo 多明我 O.P., 1170–1221) being the second (十五端經第一立者，是真天主降生之聖母瑪利亞。……第二立十五端經者，乃聖父多明我) (del Rosario 2013, vol. 2, ff. 17–18, pp. 118–19). The early Chinese texts relating to the Rosary by the Jesuits usually do not mention St. Dominic by name. Only in the *Life of the Holy Mother* (*Shengmu xingshi* 聖母行實, Vita Maria, 1631) had Alfonso Vagnone (Wang Fengsu 王豐肅/Gao Yizhi 高一志S.J., 1568–1640) mentioned the saint a few times. For example, St. Dominic exhorted a man to recite the Holy Mother's fifteen points (一士人某……幸逢名聖多敏我以善勸，令執聖母十五端課，免生前死後大難) (Vagnone 1984, vol. 3, f. 10, p. 1439). While in *The True Peace of Humankind*, del Rosario promoted the intercession of the founder of the Order of Preachers as well as Mary's. St. Dominic was appointed the second founder of the Rosary during a Marian apparition. Once, St. Dominic was praying for the sinners, and Mary appeared to him, saying, "My beloved son Dominic, you should know that the Trinitarian Lord of Heaven solely chooses my prayer of the fifteen points as the tool to remove sins from people. If you want the heterodox people to convert, you can use this prayer (我所爱兒子多明我，爾當知三位一體天主，要除人之罪，獨選我十五端經器具。爾欲異端人改變，可用此經)." Then, Mary taught the rules of the prayer and ordered St. Dominic to spare no effort to propagate the prayer (del Rosario 2013, vol. 2, f. 19, pp. 121–22).

Then, the second volume explains what the systematic counting of the Rosary signifies. The fifteen points with fifteen Our Fathers firstly signify the fifteen virtues (*shande* 善德), which people may obtain through reciting the Rosary (del Rosario 2013, vol. 2, f. 30, p. 143). The fifteen virtues are listed as follows: faith (*xin* 信), hope (*wang* 望), love (*ai* 愛), religion (*qin* 欽), practical wisdom (*zhi* 智), justice (*yi* 義), courage (*yong* 勇), penance (*lian* 廉), and the seven virtues opposite the seven sins (*ke qizui zhe qide* 克七罪者七德). In the early 1610s, Diego de Pantoja (Pang Diwo 龐迪我 S.J., 1571–1618) wrote *Seven Victories* (*Septem Victoriis, Qike* 七克) to expound the seven deadly sins and the seven capital virtues. De Pantoja's work was soon spread widely among Christian communities and became known to non-Catholics as well (Chen 2018, pp. 209, 215). The reason why del Rosario did not explain the seven deadly sins (pride, greed, lust, envy, gluttony, wrath, and sloth) and the seven capital virtues (chastity, temperance, charity, diligence, patience, kindness, and humility) was probably that his manuscript aims to serve Christian communities who would already be familiar with the schema. Also, according to St. Bernard of Clairvaux (O. Cist., ca. 1090–1153), if one recites fifteen Our Fathers every day, the total number of the recitation in a year matches the number of the wounds Jesus suffered during the Crucifixion. As an influential promoter of Marian devotion, St. Bernard of Clairvaux is constantly quoted in the manuscript. Finally, human beings live in the eleven celestial spheres (*shiyi chong tian* 十一重天) and there are four elements (*si yuanxing* 四元行). Del Rosario simply connected these fifteen points to the Our Fathers and never explained either the underlying medieval cosmology or the Aristotelian categories. He probably considered this unnecessary since the Jesuits had published several works discussing the two themes since the early seventeenth century, such as *Cosmological Epitome* (*Qiankun tiyi* 乾坤體義, 1608) written by Ricci, *Explanation on the Great Being* (*De Caelo, Huanyou quan* 寰有詮, 1628) co-translated by Francisco Furtado (Fu Fanji 傅汎際 S.J., 1589–1653) and Leo Li Zhizao 李之藻 (1565–1630), and *Investigation of the Celestial Phenomena* (*Kongji gezhi* 空際格致, ca. 1633) written by Vagnone.

The 150 Hail Marys firstly signify the 150 psalms; secondly, there is the deluge sent by the Lord of Heaven that lasted for 150 days; thirdly, there is Mary's 150 joys received from her conception, 150 sufferings from the Crucifixion of Jesus, and 150 glories from the Resurrection of Jesus (del Rosario 2013, vol. 2, f. 31, p. 144). The second volume continues to explain what the three strings, the five decades in each string, the fifty Hail Marys in each string, and the ten Hail Marys in each decade signify. The systematic counting of the Rosary is a typical late medieval devotional practice, which is believed to improve the efficacy obtained by reciting the prayer (Brown 2007, pp. 195–97, 223–25).

After confirming the efficacy of Mary's intercession, del Rosario exhorted Chinese Catholics to recite the Rosary both individually and collectively. The third volume of *The True Peace of Humankind* contains two chapters on the Holy Congregation of the Fifteen Points (*shiwu duan shenghui* 十五端聖會), a translation of rules of the Confraternity of the Holy Rosary, established by St. Dominic. Once Mary saved St. Dominic and a group of pirates at sea, the saint then founded the Holy Congregation of the Fifteen Points (del Rosario 2013, vol. 3, pp. 179–82). The fifth volume of the manuscript contains a number of miracle stories relating to this congregation (del Rosario 2013, vol. 5, pp. 238, 243, 245–52, 256–57). The congregation members shall recite the fifteen points in communion with each other and pray for the deceased (del Rosario 2013, vol. 3, f. 50, p. 182). In the sixteenth and seventeenth centuries, devotion to Our Lady of the Rosary became popular in Europe and was often associated with the Rosary confraternities (Luria 2001, p. 119). Del Rosario introduced the congregation rules and spiritual graces, revealing his intention to found the confraternity of the Rosary in local Christian communities in Fujian.

From 1676 to 1700, eighteen new Dominicans entered mainland China (von Collani 2001a, p. 324), including del Rosario. In the summer of 1676, del Rosario entered Zhangzhou 漳州 and was in contact with local Chinese Catholics, such as the literatus Paulus Yan Mo 嚴謨 (ca. 1640–post 1718) (Menegon 2009, p. 298, note 54). Yan Mo was a leader of the local Christian community and an active supporter of the Jesuit strategy of adaptation during the Chinese Rites Controversy (Yang 2021, pp. 89–90). From the 1680s to the 1690s, Yan Mo wrote several works and letters to oppose the Dominican's negative attitude toward traditional Chinese rites. Consequently, the vicar apostolic of Fujian Magino Ventallol (Ma Xinuo 馬熹諾 O.P., 1647–1732) deprived Yan Mo and his community of sacraments in 1695 (Menegon 2009, pp. 235–36, 289). Despite all the difficult and even painful experiences caused by the Dominicans, Yan Mo appreciated the newly arrived missionary del Rosario. At the latter's request, the former meticulously polished *The True Peace of Humankind*. Unfortunately, in 1686, del Rosario passed away at the age of 45, leaving the manuscript unpublished. Since, according to Yan Mo, del Rosario carried the manuscript to the mountains in Luoyuan 羅源 in the summer shortly before his death (Yan 2013, ff. 3–4, pp. 89–90), the manuscript was probably finished in the missionary's last few years. Ten years after del Rosario's death, Yan Mo wrote a preface for the manuscript, earnestly calling for its publication. At the beginning of the preface, Yan Mo cited the angels' praise on the night of Jesus' nativity (Luke 2: 14) to explain the title of the manuscript: the true peace of humankind comes from Jesus and Mary.

> "Glory to God in the highest forever, and peace to good people below forever." Such are the words in Heaven (*tianshang yu* 天上語) not the words on the Earth (*shijian yu* 世間語). If you want to search for the peace of Earth, from whom can you find it other than Jesus and the Holy Mother?

> "上願天主永永榮福，下願善人永永和平。"此天上語，非世間語也。然則欲求世之和平，豈有外於耶穌與聖母哉？ (Yan 2013, f. 1, p. 85)

Yan Mo confirmed Mary's intercession together with that of Jesus; then, he recounted how Master Ou (*Oushi* 歐師), i.e., del Rosario, arrived in Zhangzhou and evangelized local people. In fact, the preface can be seen as a brief biography of del Rosario, focusing on this Dominican missionary's ardent propagation of the Rosary.

> [Master Ou] also loved the Holy Mother to the end. At the end of the homily on the worship day, he would always conclude by talking about the prayer of the fifteen points. In addition, he prayed for the realization of one wish of Mo in exchange for reciting the prayer of the fifteen points all his life.

> 又愛聖母極切，凡瞻礼日講道尾後，必歸講於十五端經。又曾易謨一願以終身誦十五端經。 (Yan 2013, ff. 2–3, pp. 88–89)

At the end of the preface, Yan Mo recounted how he was asked by some Church leaders to revise the manuscript in 1696 (the year *Bingzi* 丙子) and longed for the publication

and circulation of Master Ou's work (Yan 2013, f. 4, pp. 90–91). The preface perfectly exemplifies that *The True Peace of Humankind*, as with most of the well-written Chinese Christian texts by the missionaries in the late Ming and early Qing, is a beautiful textual collaboration between the missionary and the Chinese Catholic.

In the early eighteenth century, Ventallol founded the Congregation of the Rose Prayer (*Meigui jing hui* 玫瑰經會) in Houban 後坂, a village near Zhangzhou (Menegon 2009, p. 256). The prescription of this confraternity of the Rosary complies with del Rosario's.[12] In 1740, the bishop of Fujian Pedro Sanz (Bai Duolu 白多祿 O.P., 1680–1747) finally published *The True Peace of Humankind* in Fu'an 福安, which circulated in Fujian at least until the 1860s (Menegon 2009, pp. 297–98). During the persecution of 1746 in Fu'an, the book was confiscated by local authorities and was recorded in the official documents, with del Rosario's and Yan Mo's names notified (Wu and Han 2008, pp. 94, 99, 104, 109, 130). *The True Peace of Humankind* must have assisted the development of local confraternities of the Rosary.

On the one hand, in terms of language, del Rosario inherited the translation from the previous manuals of the Rosary, including terms coined by the Jesuits and usages borrowed from Buddhism. Following Yan Mo's revisions, the manuscript is well-written with less awkward transliteration than the previous manuals. On the other hand, in terms of content, this comprehensive manuscript strictly conforms to the Catholic tradition in Europe, therefore with less adaptation to Chinese culture. One striking difference from the previous manuals is that the manuscript does not emphasize the notion of filial piety (*xiaodao* 孝道), an essential Chinese traditional virtue. Since Ricci's time, the Jesuits connected filial piety to Catholic devotions (Yang 2021, pp. 25–28). In the *Regulation of Reciting the Beads* (ca. 1619) attributed to João da Rocha (Luo Ruwang 羅儒望 S.J., 1565–1623), the petition of the fifth joyful mystery prays to become filially pious (*chengxiao* 成孝) (da Rocha 2002, f. 12, p. 538). Moreover, the petition of the fourth sorrowful mystery prays: "Also, like carrying one heavy and large Cross by the body, grant me the ability to often take up the Cross of holy filial piety" (亦如身負重大的十字架一般，又賜我能勤荷聖孝之十字) (da Rocha 2002, f. 20, p. 554). In a miracle story recorded in the *Life of the Holy Mother*, a widow asks her daughter to serve the Holy Mother with filial piety (*xiaojing* 孝敬) (Vagnone 1984, vol. 3, f. 56, p. 1531). In the lengthy manuscript, however, the character "*xiao* 孝" is only briefly mentioned once as a virtue among many in the general introduction of praying (del Rosario 2013, vol. 1, f. 5, p. 93).

## 4. Highlighting Mary's Intercession through Analogy with Flowers

The early manuals of the Rosary in Chinese had already mentioned that one remarkable reason to recite the Rosary is to gain merits through Mary's intercession. As the publication of Chinese Christian texts increased, missionaries expounded and categorized various merits gained by reciting the Rosary. On 31 July 1679, Pope Innocent XI (1611–1689, papacy: 1676–1689) confirmed the indulgences granted to the Confraternity of the Holy Rosary.[13] An early Chinese translation of Innocent XI's confirmation of the indulgences was written on a one-folio sheet and was undated and anonymous.[14]

> For all who are in the Teaching to gain indulgence (*dashe* 大赦), with the reciting beads, one holy medal, or one holy image, often recite Our Father thirty-three times, Hail Mary sixty-three times, or an entire string of the Rose Prayer…

> 一凡在教者得大赦，或在念珠，或在聖牌、聖像等一件，而常誦天主經三十三遍，聖母經六十三遍，或玫瑰經全串 (Anonymous 2009, p. 87)

The text lists three rosaries, i.e., the Reciting Beads for Our Lord, the Reciting Beads for the Holy Mother, and the entire Rosary. Together with the universal Church, local Christian communities across China were encouraged to recite various rosaries. In addition to spiritual benefits, such as virtues, the absolution for sins, and indulgence, reciting the Rosary may also gain other merits through Mary's intercession.

Early in the *Life of the Holy Mother* by Vagnone, the chapter "The Holy Mother Blesses Those Who Recite Prayers Such as the Rosary Beads" (*Shengmu tiyou song meigui zhu deng*

*jing zhe* 聖母提佑誦玫瑰珠等經者) records a number of miracle stories recounting various merits gained by reciting the Rosary. Uncategorized, these miracle stories mostly recount spiritual merits, such as the absolution of sins, and occasionally recount other merits, such as recovery from illness and delivery from drowning. Different from the previous Chinese texts about the Rosary by the Jesuits, *The True Peace of Humankind* presented Mary's intercession in miracle stories with a more organized structure. The manuscript records two categories of miracle stories, each with five subcategories: 1. miracle stories about saving the soul and recounting merits, comprising the delivery from Satan, from committing sins, and from the hell, the absolution of sins, and various blessings (in the fourth volume), and 2. miracle stories about saving the body and recounting merits, comprising the delivery from death, from illness, from poverty, and from various disasters, the granting and the resurrection of the son (in the fifth volume). The vivid miracle stories aimed to make Mary's intercession more appealing as well as more convincing.

It is noteworthy that the manuscript visualizes the efficacy of Mary's intercession through an analogy with flowers. Del Rosario featured the analogy in the specific petition to Mary at the end of each part of the Rosary. At the end of the joyful points:

Now with a humble heart, I present the five joyful points as one new flower to you and beg you to pray for me to your beloved Son…

我今以歡喜五端，新花一朵，謙心獻爾求爾，代我祈爾所爱之子 (del Rosario 2013, vol. 3, f. 38, p. 159)

At the end of the sorrowful points:

These five points as one red flower are colored by the blood shed by the good Lamb. Now I pray that you pray for me to your Son Jesus…

此五端紅花一朵，乃以良善羔羊所流之血綵成者。我今祈爾代我轉祈爾子耶穌 (del Rosario 2013, vol. 3, f. 43, p. 168)

At the end of the glorious points:

These five points as one white flower decorate the crown of beautiful jade. Now with a whole and humble heart, I present this crown to you and beg you to grant the crown of beatitude to me and to all the people who recite your fifteen points. Amen.

此五端白花一朵，裝成美玉之冠。我謙心切意送獻爾求爾，亦以真福之冠賜我，及凡念爾十五端之人。亞孟。 (del Rosario 2013, vol. 3, f. 47, p. 177)

The analogy with flowers in Marian prayers derives from the Catholic tradition in Europe. As a most appreciated flower, the rose became a prominent symbol of Mary. In the early thirteenth century, the writer Gautier de Coincy (1177–1236) praised Mary as a flower, especially as a rose, in various chansons, such as "She is the flower, she is the rose", "Rose of the roses", and "Flower of all goodness" (Cohen 1952, p. 23). In the same century, the hat fashioned with flowers became a luxury gift in Paris. In 1343, one Flemish beguinage rule described their daily Marian recitation as a hat of flowers. In 1458, the beguinages in Lille began to put a hat of roses or other flowers in season on the head of Our Lady (Viller et al. 1987, p. 942). Gradually, the Catholic Church referred to the flower crown as the rosaries.

Echoing the reciting rules of the third volume, the analogy with flowers appears in several miracle stories to highlight the efficacy of Mary's intercession. In the second chapter of the fifth volume of *The True Peace of Humankind*, "The Prayer of The Fifteen Points Saves Human Body from Illness" (*shiwu duan jing jiu ren roushen yu bing* 十五端經救人肉身於病), one miracle story recounts that a good man who recited the prayer of the fifteen points every day became ill. His mouth and lips became swollen and rotten, giving off an extreme odor. With extreme patience and humility, the man recited the Rosary unceasingly and had a vision.

At that time, an angel descended to bring him to visit the place where the Lord of Heaven prepares the rewards. At once, the man saw an extremely glorious view where stood an extremely beautiful rose bush. The rose bush produced one hundred and fifty white flowers, which were divided by fifteen red flowers, such is the image of the fifteen points. Then, the angel introduced the man to an extremely gorgeous palace, where everything was made of extremely shining jewelry. The Holy Mother was in the center, surrounded by many angels. The Holy Mother shined, brighter than the sun and the moon. She leaned towards them to hold the patient and spread her milk on his mouth and lips, saying: "These lips may praise me permanently, how can they be rotten and stink like this?" Then, she said to the patient: "My beloved son, now your mouth is healed and will be always fragrant. You shall praise me with your mouth." After saying these words, the Holy Mother disappeared. The patient was healed, and he exhorted people to recite the prayer of the fifteen points for the rest of his life.

時一天神降臨攜之同行，使見天主所備將賞之所，即見一極光榮美景中有一株極美玫瑰花，發一百五十蕊白花，間以十五蕊紅花，乃十五端之像。又再引見一極麗殿，皆極光寶石所造。聖母位在中，許多天神環衛。聖母顯光，超過日月，遂俯抱病人，而以己乳敷其口唇，云："此唇許長久稱讚我，何當爛臭如此？"復對云："我所愛之子，爾今已愈爾口，以後悉香。自爾口以稱讚我。"聖母言畢不見。病人愈，後平生廣勸人念十五端經。 (del Rosario 2013, vol. 5, ff. 80–81, pp. 243–44)

One entire string of the Rosary contains one hundred and fifty times of Hail Mary and fifteen times of Our Father. The rose bush with white and red flowers is a vivid image of the recitation. Mary treated the patient as a caring mother to her child. This miracle story uses the character "*ji* 極" (extremely) many times to underline the efficacy of reciting the Rosary.

In the fifth chapter of the fifth volume, "The Prayer of The Fifteen Points Saves Human Body from Disasters" (*shiwu duan jing jiu ren roushen yu zhuhuo* 十五端經救人肉身於諸禍), the last miracle story is about a young man who recited a string of five points every day. Once on a trip, he went into a chapel to recite a string of the five points, unaware that several robbers had taken his horse and were waiting outside the chapel to kill him.

The robbers waited, yet the young man did not come out. They peeked into the chapel and saw that the young man was kneeling before an extremely solemn woman. Then, they saw roses coming from the young man's mouth constantly. Whenever one rose came out, the woman took it and gave it to her son. Her son made a beautiful crown from the rose and put it on the woman's head. In fact, the rose is the prayer of the fifteen points, the woman is the Holy Mother, and her son is Jesus. When the young man stood up, both the woman and her son disappeared. On seeing this, the robbers knelt down and put their heads on the ground, begging the young man to absolve their sins. After telling him what they had seen, the robbers immediately converted from evil to good, and they honored and served the Holy Mother with extreme earnestness.

又等不出，向堂窺之，見其在極莊嚴女人前跪，又見其口中屢出一蕊玫瑰花。每一花出，女人悉取以與其子。其子將花成一美冠，送戴女人頭上。盖花者十五端經，女人即聖母，其子耶穌也。後生起，女人與其子皆不見。強盜見此事，即跪叩頭，求後生赦其罪，對說所見之事，亦遂改惡遷善，極切奉事聖母。 (del Rosario 2013, vol. 5, f. 88, p. 259)

The efficacy of reciting the Rosary is visualized as roses coming out from the young man's mouth one by one. Mary handing the rose to Child Jesus symbolizes her intercession; Jesus making the crown symbolizes his acknowledgement of the praying from the young man and Mary; and Jesus putting the crown on Mary's head symbolizes her superior status as the Holy Mother.

Almost half a century prior, Vagnone had also recounted the visualization of flowers coming out of the mouth in the *Life of the Holy Mother*. A virtuous man was ordered by the Holy Mother to recite Hail Mary sixty-three times every day. During the recitation, his mouth was giving out beautiful flowers; then, an angel would make the flowers into a crown and put it on the virtuous man's head (Vagnone 1984, vol. 3, ff. 10–11, pp. 1439–41). This miracle story promotes the Rosary which Longobardo translated as the Reciting Beads for the Holy Mother (*shengmu nianzhu* 聖母念珠). The merits of reciting the Rosary are underlined by the visualization of flowers coming out of the mouth and the making of a flower crown.[15]

Both Vagnone's and del Rosario's stories derive from a popular medieval miracle story called "The Legend of the Knight and the Rosary of 50 *Ave*". The knight had a habit of putting a flower crown on the head of the statue of Mary every day. Later, he entered a monastery and had no time to pick flowers to make the crown. An old monk suggested he recite Hail Mary 50 times a day as a flower crown, and he followed this suggestion. Once during a long trip, the young monk got down from his horse to recite his daily prayer. A robber took his horse and wanted to kill him, but he was astonished by what he saw.

> "A very beautiful lady was holding a strip used for making a crown. Every time the monk recited Hail Mary, the lady picked from his lips a rose which she attached to the strip. When a crown of 50 roses was made, she put the crown on her head and disappeared. The robber approached to the monk and interrogated him about the lady. The monk told what he did but assured that he saw nothing. The robber understood that the lady might be Our Lady so he gave back everything to the monk". (Gourdel 1952, p. 660)

The story dates back to the thirteenth century, and the Camaldolese abbot P. Sylvain Razzi (E.C.M.C., 1527–1621) said that the monk was Carthusian (Gourdel 1952, p. 661). The story became a prototype for miracle stories that propagate various rosaries. Following this tradition, the Jesuits and the Dominicans used the story to propagate the Rosary in China.

Around 1702, Pedro de la Piñuela (Shi Duolu 石鐸琭 O.F.M., 1650–1704) published the *Prayer of The Holy Mother's Flower Crown* (*Shengmu huaguan jing* 聖母花冠經) to propagate the Franciscan Crown Rosary.[16] In the seventeenth and eighteenth centuries, only a few Franciscans and Dominicans were capable of writing works in Chinese, and de la Piñuela is a remarkable example (Hsia 2011, pp. 225–26). This excellent scholar wrote several works in Chinese (von Collani 2001b, p. 333). He attributed the related miracle story to Saint Giovanni da Capestrano (Ruohan Jia-bi-si-da-nuo 若翰嘉俾斯大諾 O.F.M., 1386–1456) (de la Piñuela ca. 1702, f. 1), who had been recently canonized in 1690. As the Franciscan saint recounted, a young man entered the Franciscan Order of Friars Minor, and the Holy Mother taught him to recite the Prayer of the Flower Crown. When the young monk was reciting the prayer, the superior saw an angel using a golden string to weave the flowers together. This included one golden flower for one Our Father and ten silver flowers for ten Hail Marys (de la Piñuela ca. 1702, f. 2).[17] The first half of this miracle story resembles Vagnone's; however, the mention of flowers coming out of the mouth is absent. In Chinese society, the missionaries used the analogy with flowers to propagate various rosaries with a few different details.

Chinese Buddhist texts also use the analogy with flowers to propagate the sutra recitation. A lotus coming out of the mouth (*kou tu lianhua* 口吐蓮花) is a rather common expression to depict various Buddhist deities, virtuous monks, or other figures.[18] In other depictions from Chinese Buddhist texts, although flowers do not come from the mouth, they do visualize the efficacy of reciting the sutra. For instance, the monk Fa'an who lived in the Sui dynasty (581–618) often recited the *Lotus Sutra* (*Fahua jing* 法華經, *Saddharma-pundarika sutra*); when he explained the sutra more than forty times, lotus-like flowers suddenly blossomed around his seat (長誦法華，講四十餘遍，忽於講坐四隅，生花一叢，有十餘枝，黃白相間，狀似蓮華) (Zhou 2023, p. 74a2–4 // R134, p. 919b14–16 // Z 2B:7, p. 460b14–16). For another instance, the monk Huigong 慧恭 who lived during the Northern Zhou Dynasty (557–581) once met his old comrade, the monk Huiyuan 慧遠, after departing for

more than thirty years. During their talk, Huigong humbly said that he was only capable of reciting the one-volume *Avalokitêśvara Sutra* (*Guanshiyin jing* 觀世音經). Huiyuan despised Huigong because this sutra was so short that even little children could recite it[19]. To Huiyuan's surprise, a miraculous vision took place when Huigong recited the Avalokitêśvara Sutra.

> Gong started to raise his voice to chant the sutra title. The amazing fragrance spread all over the house. When he entered the content, music was playing from Heaven and four kinds of flowers were raining down.

> 恭始發聲唱經題，異香氛氳遍滿房宇。及入文，天上作樂，雨四種花。 ([Shi 2023](#), p. 687a8–10)

In the late Ming dynasty, Xu Changzhi 徐昌治 (1582–1672), compiler of the anti-Catholic work *Anthology of Destroying the Pernicious* (*Poxie ji* 破邪集, 1640), included Huigong's story in his compilation *Selection from Biographies of Eminent Monks* (*Gaoseng zhaiyao* 高僧摘要, ca. 1644–1672) ([Xu 2023](#), p. 285b24–c23 // R148, pp. 665b14–666b01 // Z 2B:21, p. 333b14–1). Both Buddhists and Catholics have used the analogy with flowers to propagate the prayer recitation in Chinese society, the former with a predilection for lotus, the latter rose. The analogy with flowers is but one specific example of how much the Catholic propagation of prayer recitation resembles the Buddhist one. Furthermore, the word "merit" (*gong* 功, *gongde* 功德), which is common in Chinese Christian texts about the Rosary, is borrowed from Buddhist texts ([Yang 2021](#), pp. 108–11). In addition, the amazing fragrance mentioned in Huigong's story resembles the fragrance of Marian devotion, which is a quality shared between flowers and Mary ([Fulton 2004](#), p. 8). Buddhism had widely spread this crucial notion among Chinese people. One may say that the propagation of the Buddhist sutra laid the foundation for the propagation of the Rosary in Chinese society. For both Buddhism and Catholicism in Chinese society, the spread of miracle stories is an effective way to attract people. The abundant miracle stories in *The True Peace of Humankind* made the book popular among Chinese Christians in Fujian, yet in the nineteenth century, Bishop Tommaso Maria Gentili (Li Hongzhi 李宏治 O.P., 1828–1888) had the woodblocks of the book destroyed for he found the stories superstitious ([Menegon 2009](#), p. 298, note 54).

## 5. Conclusions

The propagation of the Rosary in China belonged to a universal campaign of the Marian devotion supported by the Catholic Reformation. The Catholic reformers propagated the Rosary in order to shift the faithful's focus from local saints to Mary and to place uniformity over local diversity ([Luria 2001](#), p. 122). In Chinese society, meanwhile, reciting the Rosary was a new practice in fierce competition with other prayer recitations, such as those of Buddhists and popular religions. Collaborating with Chinese Catholics, the missionaries of different orders published various Chinese texts to propagate the Rosary. Based on the manuals written by the Jesuits, the Dominican missionary del Rosario wrote the first full-length book on the Rosary in Chinese. With its comprehensive content, especially the well-categorized miracle stories, *The True Peace of Humankind* enriched Chinese Christian texts about the Rosary and assisted propagation.

Nevertheless, while written in better Chinese, del Rosario's manuscript in fact seems to be more "European" than the previous manuals written by the Jesuits. The manuscript indicates the general attitude of the mendicants in the Chinese Rites Controversy. Contrary to the mild attitude of the Jesuits, the mendicants tried to train Chinese converts to live a full, zealous, and faithful Christian life as in Europe ([Bürkler 1942](#), pp. 21–22). On the one hand, del Rosario respected Chinese culture to the extent that he succeeded in writing a lengthy manuscript in Chinese. Furthermore, the fact that Yan Mo revised the manuscript and wrote an ardent preface indicates that this Dominican missionary might have held a positive attitude toward traditional Chinese rites. On the other hand, del Rosario conformed to the Catholic tradition in Europe to the extent that his manuscript contained less adaptation to Chinese culture. Such a contradictory characteristic of the manuscript im-

pels us to reflect on the ever-intriguing question: to what extent can Christianity remove its European post-Renaissance shell and adapt to other cultures' thought patterns, norms, and ritual lore without losing its identity (Zürcher 1994, p. 63)?

Since the late Ming and early Qing, Chinese Catholics have embraced the efficacy of Mary's intercession and favored reciting the Rosary. Yet, the non-Jesuit texts about the Rosary such as *The True Peace of Humankind* written by a Dominican and the *Prayer of The Holy Mother's Flower Crown* written by a Franciscan await more research. It is noteworthy to examine the third area of questions mention in the Introduction. In addition to the polemic on traditional Chinese sacrificial rituals, contextual studies on the texts relating to Catholic devotions provide a new perspective to investigate the Chinese Rites Controversy. Moreover, it will be both challenging and rewarding to further examine the propagation of the Rosary in Chinese society in comparison with the propagation of the Buddhist sutra.

**Funding:** This research received no external funding.

**Data Availability Statement:** No new data were created or analyzed in this study.

**Acknowledgments:** Thierry Meynard, S.J., from Sun Yat-sen University offered invaluable suggestions to my article. Zurong Yang 楊祖榮 from Fujian Normal University kindly assisted me with understanding Chinese Buddhist texts. Two anonymous reviewers offered enlightening comments. Brendan Gottschall, S.J., from Boston College has done a meticulous proofreading. I am deeply indebted to each of them.

**Conflicts of Interest:** The author declares no conflicts of interest.

## Notes

1. Fu Ji conjectures that the Jesuits might mistake the character "端 *duan*" for "段 *duan*" (Ji 2016, p. 41, note 76). As a matter of fact, "*duan* 端" is a proper translation for a decade (*decadem*, set of ten), i.e., ten beads of the chaplet.

2. According to Henri Bernard, the first edition of the *Daily Services of the Holy Teaching* was published as early as in 1602 (Bernard 1945, part 3, p. 321). Paul Brunner, on the other hand, presumes that it was published in 1619 (Brunner 1964, p. 27). This prayer collection has been re-edited and republished over time. As mentioned above, Ferreira's manual of the Rosary was included in the 1665 edition.

3. According to the Chinese Christian Texts Database, the literatus Zhang Geng 張賡 (ca. 1570–1646/47), the collaborator of Aleni's *Images of Fifteen Points*, also compiled a text called the *Images and Prayer of Fifteen Points in the Rose Prayer* (*Meigui jing shiwu duan tujing* 玫瑰經十五端圖經, ca. 1637). https://libis.be/pa_cct/index.php/Detail/objects/1492 (accessed on 7 October 2023). Both texts are housed in St. Petersburg, Vakhtin-Katalog fonda kitajskich ksilografov Instituta vostokovedenija AN SSSR, shelf: D, 202. no. 3171.

4. For a brief introduction to *The True Peace of Humankind*, consult the Chinese Christian Texts Database, available online: https://libis.be/pa_cct/index.php/Detail/objects/10025 (accessed on 7 October 2023).

5. For a different translation in the early manual by the Jesuits, consult the manual attributed to João da Rocha (da Rocha 2002, ff. 1–2, pp. 516–18).

6. In 2002, Pope John Paul II (1920–2005) added the five Luminous mysteries to the Rosary.

7. On 1 November 1950, Pius XII (1876–1958) defined the Assumption of Mary as a dogma. It is noteworthy that the expression "to ascend into the Heaven" (*shengtian* 升天) is used for both the Ascension of Jesus and the Assumption of Mary in Chinese.

8. Paul Brunner also used the present tense in his translation in French (Brunner 1964, pp. 191–98).

9. Consult Paul Brunner's opinion of the Litany of Chronicle of Jesus' Life and Sufferings (*Lie xu Yesu xingji kunan daowen* 列敘耶穌行蹟苦難禱文) and the Litany of the Holy Mother's Life (*Shengmu xingji daowen* 聖母行蹟禱文) (Brunner 1964, pp. 46–49). In addition, Brunner pointed out that these two litanies provided Chinese Catholics with "a useful method to vary and to revivify the recitation of the Rosary" (Brunner 1964, p. 51).

10. For the day of worship and the seven days in a week, consult Yang's work (Yang 2021, pp. 229–31).

11. As recorded in *Sequel to Li Jiubiao's Diary of Oral Admonitions* (*Xu Kouduo richao* 續口鐸日抄, ca. 1698), members of the Confraternity of the Holy Mother in Shanghai cared about which points of the Rosary should be recited according to the Catholic liturgical calendar. Those who recited the five joyful points during the Lenten season were scolded (Zhao 2007, pp. 611–12).

12. For the prescription of the Congregation of the Rose Prayer, consult Eugenio Menegon's work (Menegon 2009, pp. 256–57).

13. For the Papal Brief (*Nuper pro parte dilecti filii Antonii de Montoy*), consult the bibliography of teachings of the Popes and Councils on the Blessed Virgin Mary, University of Dayton, available online: https://udayton.edu/imri/mary/c/church-teachings-in-the-early-modern-1600-1800-period.php (accessed on 7 October 2023).

14    According to the Chinese Christian Texts Database, the Chinese version (BnF, Chinois, 7276–II) was written before 1750. https://libis.be/pa_cct/index.php/Detail/objects/1647 (accessed on 7 October 2023).

15    Among the ten meanings of "*gong* 功" given by *Le Grand Ricci*, the first one is of merit. The meaning of effort and that of effect could also be applied to the term (Institut Ricci 2001, III, p. 991).

16    For the introduction of the *Prayer of The Holy Mother's Flower Crown*, consult Paul Brunner's work (Brunner 1964, p. 132).

17    Also consult *Rule of the Third Order of St. Francis* by de la Piñuela (de la Piñuela 2009, p. 191).

18    "口吐蓮華" (Bodhiruci 2023, p. 346a26); "口出蓮華蒲桃朵葉" (Bodhiruci 2023, p. 349b2); "口吐蓮花" (Chaoyong 2023, p. 427c20 // R140, p. 201a6 // Z 2B:13, p. 101a6).

19    Avalokitêśvara Sutra is also called Avalokitêśvara Sutra of King Gao (*GaoWang Guanshiyin jing* 高王觀世音經) (Liu 2008, p. 152).

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
