# Peer review of "Propagating the Rosary in the Early Qing—A Case Study of del Rosario’s Comprehensive Manuscript"

_religions, doi:10.3390/rel15020230_

Round 1

Reviewer 1 Report

Comments and Suggestions for Authors

This article presents ‘the true peace of humankind’, a manuscript written in 1680 in Chinese by missionary Arcadio del Rosario about the practice of Rosary. In a missionary context in which texts about devotional practices such as the rosary multiplied, the author argues that this text “enriched the Chinese Christian texts about the Rosary and assisted the propagation” (L.480). This paper is well structured, informative, and clearly written. It shows well the fluidity of the practices related to the rosary (L.87-99), it provides a good introduction to the topic and situates well its manuscript within its historical and religious context.

Yet, this paper needs to better highlight the specific contribution of the analyzed manuscript. While some sections read as lists of facts (L.100-118) other statements are repetitive (L.180-182, L277-280) and the argument of the paper becomes less obvious. What is the core contribution – and eventually innovation – of this manuscript about the rosary? Which aspect of the manuscript requires -according to the author- more attention to understand its specificity? What does it say about the rosary in general and the Chinese inter-religious context? I personally believe that more attention could be given to the interreligious aspect in which this manuscript emerged, but this needs to be decided based on the manuscript. The author may want to underscore other aspects. Anyhow, the author needs to highlight the core argument of this paper to better show why scholars should look at this manuscript.

A few additional comments:

·         L.31-38: I would recommend having a brief preliminary definition of a rosary for readers who might be familiar with this practice.

·         P.66-68: I wonder whether the number 15 would deserve more scrutiny and interpretation.

·         L. 219: Why “mout” Luoyuan? Today, this is a city behind a series of mountain, but it is usually not referred as “Mount Luoyuan” (like Mount Wuyi for instance). If this was the historical practive, mount can be maintained, but please double check!

·         L.404: source of this legend? Gourdel 1952?

·         L.474: In China, where Christianity had a limited footprint, there was not yet “local saints” to compete with. Rosary was used in a inter-religious context.

Author Response

First of all, I’d like to express my gratitude for your meticulous and enlightening suggestions and your warm encouragements! Please find the specific response below.

  1. For highlighting the core argument of this paper and the significance of this manuscript, cf. line 12–15, 26–29, 308–326, 556–561, 565–567.
  2. For a brief preliminary definition of a rosary, cf. line 37–38.
  3. For the interpretation of the number 15, cf. line 214–246.
  4. For a correction of “mount” Luoyuan, cf. line 266.
  5. Yes, the source of this legend is Gourdel 1952, indicated after the citation, cf. line 483.
  6. Yes indeed, there was no “local saints” in China to compete with. For a connection in the context, cf. line 549–551.

I will continue to reflect on your suggested research questions in my future research on the Rosary and other Catholic devotions in Chinese society.

Reviewer 2 Report

Comments and Suggestions for Authors

I would like to express my appreciation for your work on the article titled "Propagating the Rosary in Early Qing—A Case Study of del Rosario’s Comprehensive Manuscript", which contributes significantly to the study of religious history in China. However, I wish to draw attention to an aspect of the methodology that, in my opinion, requires additional consideration.

I have noticed that the article does not explicitly define the research objective and the research methods. Clearly articulating the research aim is crucial for understanding the scope of the work and its specific intentions. It allows the reader to better comprehend why the topic was undertaken and what the expected outcomes of the research are.

Similarly, a description of the research methodology is essential for assessing the credibility and scholarly value of the work. A detailed presentation of the methods, both in terms of sources and analysis, would allow for an assessment of how data was collected and analysed, which is crucial for evaluating the research findings. In the context of your article, such a description could elucidate how you conducted the analysis of historical texts, how you interpreted the source materials, and what criteria were applied in their selection.

I believe that incorporating these elements would not only strengthen the methodological structure of the article but also contribute to a better understanding of its significance in the context of studies on the history of religion in China.

Author Response

Thank you for your helpful suggestions and your kind appreciation for my article on the Rosary! Please find the specific response below.

  1. For better stating the research questions and methods, cf. line 12–15, 26–29, 308–326, 556–561, 565–567.

I am afraid that this revision is what I am capable to make within a limited time. I will try to define the research objective and the research methods more explicitly in my future research on the Rosary and other Catholic devotions in Chinese society.